# Improving the Potential of Coniferous Forest Aboveground Biomass Estimation by Integrating C- and L-Band SAR Data with Feature Selection and Non-Parametric Model

**Yifan Hu [1], Yonghui Nie [1], Zhihui Liu [2], Guoming Wu [3] and Wenyi Fan [1,***

[1]  Key Laboratory of Sustainable Forest Ecosystem Management, School of Forestry, Northeast Forestry University, Ministry of Education, Harbin 150040, China; yifan_hu2021@nefu.edu.cn (Y.H.); nieyonghui@nefu.edu.cn (Y.N.)

[2]  International Institute for Earth System Science, Nanjing University, Nanjing 210023, China; liuzh@nefu.edu.cn

[3]  Jiamusi Forestry and Grassland Administration, Jiamusi 154000, China; wgm@nefu.edu.cn

*   Correspondence: fanwy@nefu.edu.cn; Tel.: +86-139-4605-5384

**Abstract:** Forests play a significant role in terrestrial ecosystems by sequestering carbon, and forest biomass is a crucial indicator of carbon storage potential. However, the single-frequency SAR estimation of forest biomass often leads to saturation issues. This research aims to improve the potential for estimating forest aboveground biomass (AGB) by feature selection based on a scattering mechanism and sensitivity analysis and utilizing a non-parametric model that combines the advantage of dual-frequency SAR data. By employing GF-3 and ALOS-2 data, this study explores the scattering mechanism within a coniferous forest by using results of target decomposition and the pixel statistics method. By selecting an appropriate feature (backscatter coefficients and polarization parameters) and using stepwise regression models and a non-parametric model (the random forest adaptive genetic algorithm (RF-AGA)), the results revealed that the RF-AGA model with feature selection exhibited excellent AGB estimation performance without obvious saturation (RMSE = 10.42 t/ha, $R^2$ = 0.93, leave-one-out cross validation). The $\sigma_{HV}$, $\sigma_{VH}$, Pauli three-component decomposition, Yamaguchi three-component decomposition, and VanZyl3 component decomposition of thee C-band and $\sigma_{HV}$, $\sigma_{VH}$, $\sigma_{HH}$, Yamaguchi three-component decomposition, and VanZyl3 component decomposition of the L-band are suited for estimating the AGB of coniferous forests. Volume scattering was the dominant mechanism, followed by surface scattering, while double-bounce scattering had the smallest proportion. This study highlights the potential of investigating scattering mechanisms, sensitivity factors, and parameter selection in the C- and L-band SAR data for improved forest AGB estimation.

**Keywords:** feature selection; non-parametric model; above-ground biomass estimation; scattering mechanism; GF-3; ALOS-2





## 1. Introduction

The rise in global temperature and climate change are significantly contributed to by the consistent increase in atmospheric $CO_2$ concentration [1–3]. Forests play a crucial role in both global climate change [4] and the carbon cycle [5]. Through photosynthesis [1], the total amount of carbon sequestered in forest vegetation is approximately one-twelfth of the $CO_2$ in the atmosphere annually [6]. The majority of this carbon is stored in forest biomass. Therefore, the accurate estimation of regional forest biomass is crucial for climate change studies, the global carbon cycle analysis, and the realization of a country's commitments toward the 'carbon neutrality' goal [7]. Typically, forest biomass includes both aboveground and underground components. However, due to the challenges associated with collecting data on underground biomass in sample plots, the current forest biomass estimates primarily focus on aboveground biomass (AGB) research [8].

Currently, remote sensing technology [8] is the primary method for estimating AGB at the regional scale. Optical remote sensing and synthetic aperture radar (SAR) data are widely utilized for estimating AGB [9]. SAR technology has overcome the limitations of traditional optical remote sensing [10] and can penetrate the forest canopy, capturing information about the deeper canopy, branches, and trunks, thus providing valuable insights into the vertical structure characteristics of the forest [11]. Consequently, SAR technology offers distinct advantages over optical remote sensing by effectively improving the saturation point and potential of estimating the forest AGB.

Dual-frequency SAR holds even greater potential for estimating AGB [12], surpassing the limitations of optical remote sensing and single-frequency SAR in terms of the sensor capabilities and penetration. Studies have demonstrated that the accuracy of the backscatter coefficient retrieval of forest AGB based on multi-frequency SAR data is higher than that of single-frequency SAR data [3,13–15]. SAR polarization decomposition parameters display high sensitivity to changes in forest canopy biomass and AGB, significantly enhancing the accuracy and saturation point of biomass estimation from SAR data [16–19]. Noticeably, the dominant scattering mechanism and complementarity of dual-frequency SAR in the estimation of forest AGB have not been considered. These previous studies have not considered feature selection based on the advantages of estimating the ground AGB using dual-frequency bands of SAR data with the backscatter coefficients, polarization decomposition parameters, and scattering mechanism.

Moreover, it should be noted that the wavelengths and penetration capacities of SAR data exhibit variations across different frequency bands. Consequently, the region within the forest that interacts with different frequency SAR signals and the prevailing scattering mechanism within the forest are also subject to differentiation. Short-wave bands (K-, X-, C-band) primarily capture smaller elements about the forest canopy including leaves (needles) and small branches [12], while long-wave bands such as the L-band and P-band exhibit strong penetration capabilities, enabling them to reach the ground through the canopy. The L-band and P-band interact with different parts of the forest including the main branches of the forest canopy, trunk, and ground [20–22]. The polarization target decomposition method can describe the forest's scattering mechanism. However, there is a lack of research on the analysis of the main scattering mechanism of SAR data in different frequency bands in the coniferous forest based on the results of polarization decomposition. Since the AGB saturation point estimated by SAR data is influenced by the band, polarization mode, and forest structure [23], L-band data have emerged as the optimal choice for the accurate estimation of forest biomass [24,25]. The C-band is suitable for the estimation of the canopy biomass of a small structure (fine branches and needle) [26]. Therefore, we used ALOS-2 data in the L-band and GF-3 data in the C-band, all of which have full-polarization, combining the advantages of the C- and L-bands to improve the potential of the estimation of AGB in dual-frequency.

With the increase in the number of multi-frequency bands and SAR biomass sensitivity factors, the limited number of forest plots inevitably leads to small samples and the 'curse of dimensionality', which results in information redundancy and affects the accuracy of the model [27]. Therefore, feature selection and appropriate methods are crucial to achieve the robust and high-precision estimation of forest AGB using multi-band SAR. The stepwise regression method selects the parameters entering the model for biomass estimation effectively, outperforming unary linear models and simple logarithmic exponential models [22,28]. However, with the increase in the number of feature parameters, the stepwise regression method becomes inadequate in capturing the complex relationship between the forest aboveground biomass (AGB) and the features. Therefore, some studies have applied a non-parametric model based on machine learning algorithms such as fast iterative feature selection with K-nearest neighbors (KNN), support vector machine (SVM), artificial neural networks (ANNs), and random forest (RF) models to estimate forest AGB using multi-band spaceborne SAR data [3,15,17,29], and the estimation results are expected to outperform the multivariate linear regression model [30]. However, only relying on machine learning

algorithms for adaptive feature selection and modeling can lead to the problem of local optimal selection. This study employed feature selection based on the scattering mechanism and sensitivity analysis and utilized a non-parametric model that combines the advantage of dual-frequency SAR data. The sensitivity analysis explored the potential of the C-band and L-band SAR data for estimating the AGB in *Larix principis-rupprechtii Mayr* forests. The three main objectives of the sensitivity analysis were to investigate the sensitivity of the backscatter coefficients and polarization decomposition parameters extracted from GF-3 and ALOS-2 SAR data concerning AGB estimation, determine whether these correlations could facilitate the identification of dominant scattering mechanisms within coniferous forests, and evaluate the efficacy of GF-3 and ALOS-2 SAR data for estimating the AGB specifically in pure *Larix principis-rupprechtii Mayr* forested areas. The non-parametric model of the random forest adaptive genetic algorithm (RF-AGA) model [31] was used, which combines feature selection and modeling into a single step. By using the minimum root mean square error (RMSE) of the model to control the optimal feature selection and modeling results, this approach avoids the problem of falling into the local optimal solutions effectively.

In existing studies on estimating the forest biomass using machine learning algorithms based on dual-frequency SAR data, the analysis of the advantages, complementarity, and dominant scattering mechanisms of dual-frequency SAR in forests has been neglected. In this context, this paper aims to achieve the following objectives:

(1) Estimate the dominant scattering mechanisms of the C-band and L-band data in the coniferous forest in north China.
(2) Assess the advantages and complementarity of the C-band and L-band data in estimating the AGB.
(3) Improve the potential and saturation point of estimating the forest AGB using feature selection combined C-band and L-band dominant scattering mechanisms, advantages, and sensitivity analysis based on dual-frequency SAR data through stepwise regression models and the non-parametric RF-AGA model.

## 2. Study Area and Data

### 2.1. Study Area

The study area was the Saihanba Forest, which is located at the junction of Hebei Province and the Inner Mongolia Autonomous Region in China (117°E, 42°N). The Saihanba Forest, which is the largest plantation forest in the world [32], covers an area of 93,461 ha. The terrain and landforms are relatively complex, consisting of plateaus, undulating hills, floodplains, and mountainous areas. It represents a typical temperate forest-grassland ecotone in North China and encompasses a diverse ecosystem that combines forests, meadows, and marshes with elevation ranges from 1010 m to 1939.9 m. The complex terrain and climatic environment pose challenges for ground surveys and optical remote sensing in estimating the AGB. Therefore, microwave remote sensing is an appropriate means of estimating the AGB in this region (Figure 1).

The main tree species in the study area include Larix principis-rupprechtii Mayr, Pinus sylvestris mongolica, Betula platyphylla, Populus davidiana, and others [33]. Among them, Larix principis-rupprechtii Mayr is the dominant species in the Saihanba region [32]. Within the coverage area of the SAR image, Larix principis-rupprechtii Mayr accounts for up to 90%. Therefore, we focused on the AGB of Larix principis-rupprechtii Mayr forests as the research object.

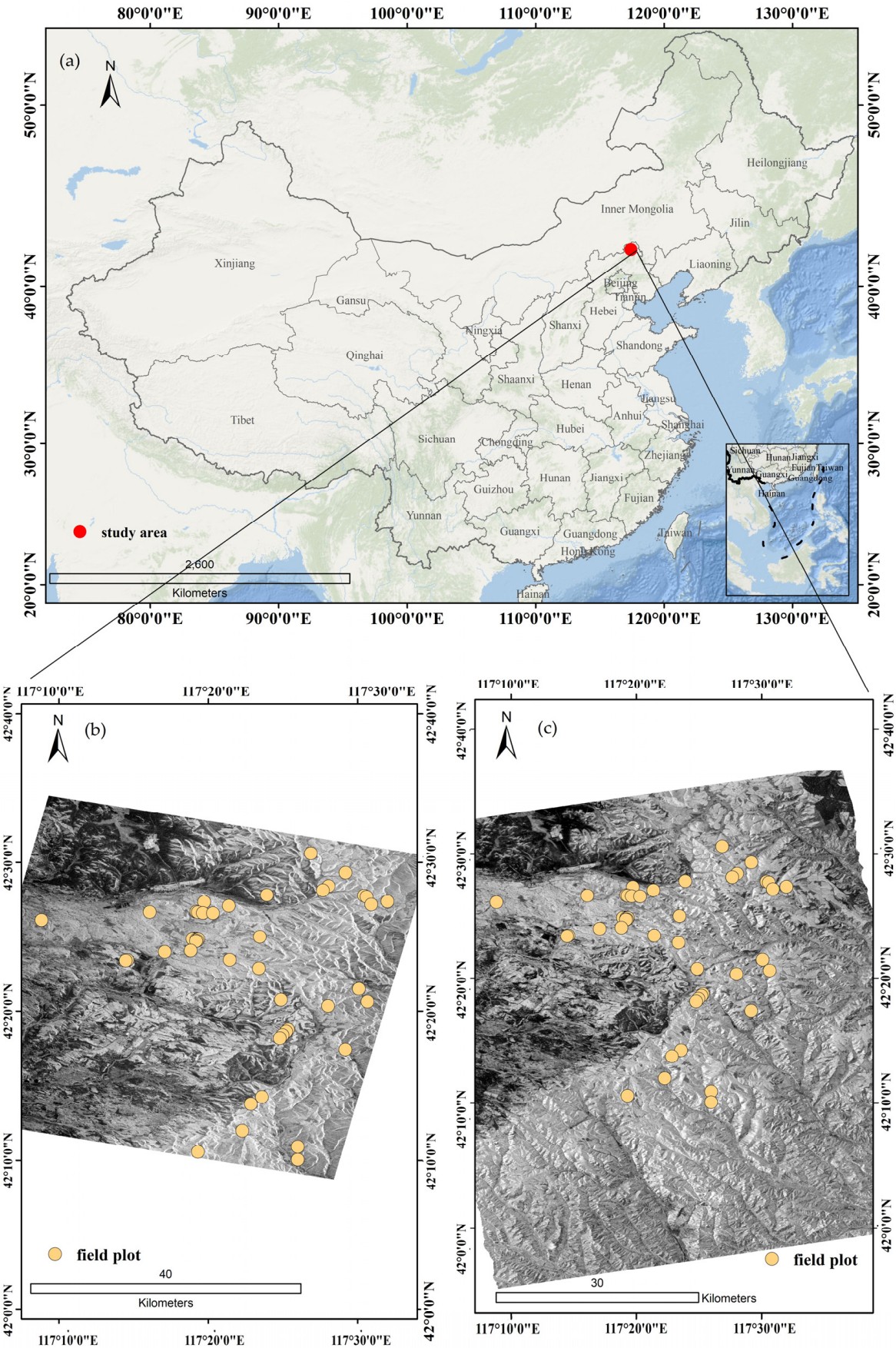

**Figure 1.** Location map of the study area. (**a**) The location of the study area; (**b**,**c**) represent 50 plots in the GF-3 and ALOS-2 images, respectively.

*2.2. Data*

2.2.1. Field Data

A total of 50 plots were established in 2021 within the SAR image coverage area of the study area. Although the plot data and ALOS-2 data as well as the plot data and GF-3 data were collected one year apart, the forest type and structure in the study area remained stable, and the change in the AGB during this period was not considered as substantial. Among them, 18 plots were located on a kilometer grid systematically, while the remaining plots were randomly sampled to represent the real distribution of the forest as much as possible. The plots were diamond-shaped with an area of 0.06 ha, and the GPS coordinates and relative positions in the stand were recorded. To avoid human disturbance, the plots were located at a distance of one times the average tree height from the forest edge. In each plot, tree measurements were conducted for trees with a diameter at breast height (DBH) greater than 5 cm using the Vertex IV instrument to measure tree height. Measurements of tree species, DBH, crown width, and tree height were recorded for live standing trees. Individual tree AGB was calculated by applying the corresponding species-specific allometric equations using tree height and DBH [34], and the values were summed to obtain the AGB of each plot. The average value of AGB measured in the plot was about 92.8 t/ha, the minimum value was 1.9 t/ha, and the maximum value was 214.3 t/ha.

2.2.2. SAR Data

SAR images from ALOS-2 PALSAR-2 in the L-band and GF-3 data in the C-band were procured, each providing full polarization channels and an image area that could fully cover the plot area.

The ALOS-2 PALSAR-2 based on the L-band image of the study region was captured on 8 August 2020. The data are of the HBQR1.1 level, with an image resolution of 2.86 m × 2.77 m, a wavelength of 23.6 cm, and an incidence angle of 27.8°.

The GF-3 SAR data of the study area were captured on 19 April 2022, which represents the fully polarized C-band data of the SCL1.1 type with Quad-Polarization Stripmap I (QPSI) as the observation mode. The image resolution was 2.24 m × 5.36 m, the wavelength was 5.5 cm, and the incidence angle was 21.3°.

2.2.3. SAR Data Processing

The SAR data processing includes radiometric calibration, speckle filtering, multi-looking, and geocoding, which can eliminate radiometric and geometric distortions [19]. The ALOS-2 and GF-3 data were pre-processed using the PolSARpro v6.0 (Polarimetric SAR data Processing and Education Toolbox 6.0), provided by the European Space Agency) and Pixel Information Expert for SAR 6.3 (PIE-SAR6.3), respectively.

All images underwent radiometric calibration to obtain the true backscattering coefficients of the reflective surface [35].

$$\sigma_{dB}^0 = 10\log_{10}\left(\left(I^2 + Q^2\right)\left(\frac{Q_V}{32,767}\right)^2\right) - K \tag{1}$$

The radiometric calibration formula for the GF-3 data is as above: I and Q denote the real and imaginary components of the SLC1.1 product, respectively. The maximum value of the original image data, denoted as $Q_V$, can be determined from the Qualify Value specified in the metadata. Additionally, K is the calibration coefficient that is also retrieved from the metadata.

$$\sigma_{dB}^0 = 10\log_{10}\left(I^2 + Q^2\right) + CF \tag{2}$$

The radiometric calibration formula for the ALOS-2 PALSAR-2 data is as follows: I and Q are the real and imaginary parts of the SLC data, and CF is the calibration coefficient [19].

Enhanced Lee filtering and multi-looking were used to reduce the coherent speckle noise in the image [36]. In this study, multi-looking with window sizes of 4 × 9 and 3 × 3 were

used for the ALOS-2 and GF-3 data, respectively. This research used ASTER GDEM V2 30 m global digital elevation data that could cover the study area to geocode the SAR images, aiming to rectify geometric distortions in the image caused by severe terrain undulations in the study area [37]. Due to the imaging characteristics of SAR data, SAR images are more susceptible to geometric errors caused by terrain changes. Therefore, this study employed the distance-Doppler algorithm for orthorectification. Additionally, the digital elevation data were utilized as auxiliary data to calculate the slope and aspect of the study area. This information, along with the local incidence angle of the SAR data, was employed to correct the SAR images. Geographic encoding was performed using the WGS84 coordinates and the UTM 50 N projection coordinate system. The geographic encoding of the ALOS-2 data and GF-3 data was carried out using the geographic encoding module within the PIE-SAR6.3 software, provided by Piesat Information Technology Co., Ltd., Beijing, China.

2.2.4. Polarimetric Parameter and Backscatter Coefficients Extraction and Combination

This study focused on the extraction of backscatter coefficients and polarization decomposition parameters from SAR images, specifically in both the C-band and L-band frequencies. The backscatter coefficients were obtained for the four polarization channels. Various operations such as difference, sum, ratio, and multiplication were performed on these four backscatter coefficients, resulting in the construction of a new combination comprising 19 parameters [17] for each frequency.

Given the complex geometry and distributed nature of forest objects, the research emphasized incoherent decomposition methods suitable for such distributed objects (except Pauli 3 component method). For the GF-3 data (C-band), decomposition methods included Freeman two-component decomposition (Freeman–Durden two-component decomposition) [38], Freeman three-component decomposition [39], AnYang three-component, AnYang four-component decomposition, Yamaguchi three-component decomposition, Yamaguchi four-component decomposition [40], Pauli three-component decomposition, and VanZyl3 component decomposition [41], and Huynen decomposition [42]. Similarly, for the ALOS-2 data (L-band), decomposition methods involve Freeman2 component, Freeman3 component, AnYang3 component, AnYang4 component, H/A/alpha eigenvalue set decomposition, H/A/alpha eigenvector set decomposition, Yamaguchi3 component, Yamaguchi4 component, and VanZyl3 component decomposition.

Furthermore, based on the obtained decomposition parameters, a series of parameter combinations were constructed. This included performing logarithmic transformations on the parameters and constructing the ratio [17] and multiplication. Previous research has demonstrated that the relationship between the ratio of decomposition parameters and forest canopy biomass exhibits a certain sensitivity [16].

In total, the number of backscatter coefficients and polarization decomposition parameters, along with their combined parameters, were 100 and 76 for the C-band and L-band, respectively.

## 3. Method

This research explored feature selection based on dominant scattering mechanisms and biomass-sensitive analysis by combining C-band and L-band data in the *Larix principis-rupprechtii Mayr* of the Saihanba Forest, North China. SAR parameters based on feature selection were applied to ground biomass modeling using stepwise regression and the non-parametric model of the random forest adaptive genetic algorithm (RF-AGA) machine learning algorithm. This study considered the impact of the sensitivity analysis and dominant scattering mechanisms on these parameters and the possibility of enhancing dual-frequency SAR biomass estimation by incorporating the advantages of the C-band and L-band.

The research process was divided into three stages: extracting SAR parameters from the remote sensing images, the feature selection dominant scattering mechanisms and biomass-sensitive analysis, and the development of AGB models (Figure 2).

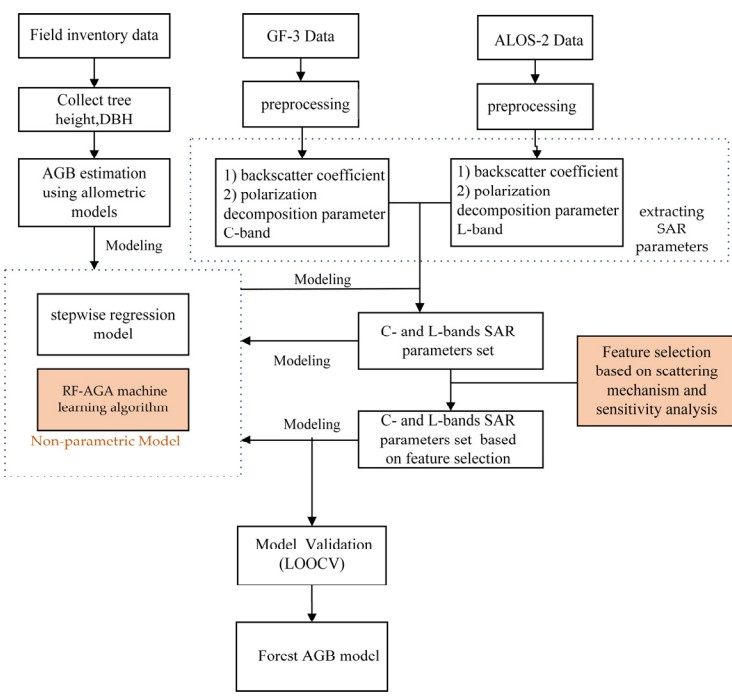

**Figure 2.** Flowchart of this study.

### 3.1. Feature Selection

3.1.1. Feature Selection Based on the Dominant Scattering Mechanism

    Polarization decomposition methods can elucidate scattering mechanisms in forests. Various polarization decomposition methods in the C-band and L-band can be used to obtain polarimetric scattering components, followed by pixel-based statistical analysis. The analysis aimed to investigate the dominant scattering mechanisms and component proportions in the *Larix principis-rupprechtii Mayr* forest based on the C- and L-bands.

    Specifically, utilizing auxiliary data from the inventory data, the total power and proportions of each physically-based polarization decomposition method within the SAR image coverage area of the *Larix principis-rupprechtii Mayr* forest were calculated (as shown in Equation (3)). This calculation helped determine the dominant scattering mechanisms of various polarization decomposition methods in the C-band and L-band within the *Larix principis-rupprechtii Mayr* forest.

$$P'_i = \frac{P_i}{P_V + P_O + P_D}(i = V, O, D) \tag{3}$$

    Here, $P_V$ represents the power of volume scattering, $P_O$ represents the power of surface scattering, and $P_D$ represents the power of double-bounce scattering. $P'_i$ denotes the ratios of volume scattering, surface scattering, and double-bounce scattering to the total power. In this study, it was assumed that $0 \leq P'_i \leq 1$ and $\sum P'_i = 1$, implying that the volume scattering, double-bounce scattering, and surface scattering components are mutually independent, and the total scattering power equals the sum of these three scattering mechanism components.

    The proportions of the scattering components for each polarization decomposition method in both the C- and L-bands were separately computed and compared to identify the similarities and differences. This analysis provides clarity on the dominant scattering mechanisms of the C-band and L-band in coniferous forests.

    In this study, a total of 100 SAR parameters (including backscatter coefficients, polarization decomposition parameters, and combinations) from the GF-3 data and 76 SAR parameters from ALOS-2 data were selected. The selection of parameters to be included in the model was guided by prior knowledge, which involved analyzing the dominant

scattering mechanisms in *Larix principis-rupprechtii Mayr* forests for the C-band and L-band frequencies as well as assessing the correlation between the SAR parameters and forest AGB. The aim was to reduce redundant parameters, streamline the selection of parameters for the model, and enhance the interpretability of the chosen parameters.

Currently, many studies have incorporated polarization decomposition parameters and backscatter coefficients directly into statistical models or a non-parametric model for biomass estimation. In this study, however, we aimed to enhance the potential of forest aboveground biomass (AGB) estimation by selecting SAR parameters based on the dominant scattering mechanisms and sensitivity analysis. After obtaining the results of target decomposition, the proportions of each scattering component were calculated. The selecting principle for the polarization scattering parameters and their combinations were as follows.

The scattering component with the highest proportion (representing the dominant mechanism) must be retained for each target decomposition method. Then, remove the scattering component parameter that accounts for less than 5%. The Helix scattering component was eliminated from the four-component target decomposition method, as complex-shaped objects that can cause left or right helix scattering are nearly absent in forests [43]. A helix scattering power term often disappears in naturally distributed scattering scenarios (e.g., forest) [40].

### 3.1.2. Feature Selection of Sensitive Factors for AGB Based on Sensitivity Analysis

Sensitivity analysis examines the relationship between variables, quantifying the association between the AGB and SAR parameters (backscatter coefficients and polarization decomposition parameters). It identifies sensitive SAR parameters for AGB. The Pearson correlation coefficient measures this relationship, with its sign indicating positive or negative correlation, and its magnitude represents the strength of association. Previous studies [44,45] have shown that applying logarithmic transformations to both backscatter coefficients and AGB before regression analysis improves the correlation coefficients and reduces average errors. Building on this, some studies have applied logarithmic transformations to the AGB values prior to calculating the correlation coefficients [45]. Following this approach, the present study applied logarithmic transformation to the AGB values for the correlation analysis.

In the C- and L-bands, the backscattering coefficients were retained for all four polarization channels (HH, VV, VH, HV) as they contain information on the interaction between the SAR and forest canopy, branches, and trunks. For the C-band, the remaining combinations of backscattering coefficients were selected from a constructed set of 15 combinations based on significance testing and an R-value greater than 0.25. Similarly, for the L-band, the remaining combinations were selected based on significance testing and an R-value greater than or equal to 0.5.

The polarization decomposition parameters and their combinations from the C-band and L-bands were individually selected according to different polarization decomposition methods. The selecting principal for parameters within each polarization decomposition method is as follows:

(1) Parameters and combinations that contain the same information are evaluated based on their correlation, and those with higher correlations are retained. This principle effectively removes redundant parameters while preserving biomass-sensitive factors.

(2) For the original polarization decomposition parameters within the same polarization decomposition method (e.g., surface scattering, double-bounce scattering, volume scattering), parameters that fail the significance analysis or have a correlation with an AGB less than 0.2 are excluded. The parameter representing the highest correlation with AGB must be retained.

*3.2. Model of Aboveground Biomass Estimation*

3.2.1. Non-Parametric Model of Adopted the Random Forest Adaptive Genetic Algorithm

The study adopted the random forest adaptive genetic algorithm (RF-AGA) model constructed by our previous study [31] to build the AGB models. The RF-AGA methodology is primarily underpinned by the framework of the genetic algorithm, which initiates the generation of a stochastic population and employs the random forest algorithm as the fitting model for the assemblage of feature subsets. The genetic algorithm emulates the double-helix coding inherent in biological chromosomal structures. Employing a binary data structure, it encodes the variable set, executes cross-mutation operations via the genetic algorithm, and subsequently recombines the structural components to preserve vital information. This elaborate sequence of procedural steps culminates in the establishment of a novel feature subset. Concurrently with its execution of feature selection, a genetic algorithm exerts control over the hyperparameters within a random forest model. The feature subsets, acquired via the genetic algorithm, are seamlessly integrated into the random forest model in conjunction with varying counts of decision trees, facilitating the computation of the root mean square error (RMSE) value. The minimal RMSE attained by the random forest regression model was designated as the target objective for the iterative progression of the genetic algorithm. This algorithm enables the simultaneous feature selection and model construction using genetic algorithms and the control of hyperparameters. By merging the two steps of feature selection and model construction into a single step, it avoids the problem of local optima and facilitates the global optimum.

This method aimed to find the optimal subset combination of features considering the model's fitting and generalization ability for high-dimensional, small-sample data. In this study, the C-band and L-band backscattering coefficients and polarimetric decomposition parameters (a total of 176 parameters, with 100 in C-band and 76 in L-band) were used together with AGB measurements from 50 sample plots to establish the AGB model. The dataset used in this study represents a typical high-dimensional, small-sample dataset that exhibits common challenges encountered when estimating the AGB using multi-frequency data. Therefore, the non-parametric RF-AGA model was employed in this study to address the issues of high-dimensionality and small-sample size associated with multi-frequency data. The objective was to improve the accuracy and saturation point of AGB estimation using dual-frequency, high-dimensional, and small-sample data.

3.2.2. Stepwise Regression Model

Increasing the model's parameters raises the complexity, computation time, and the risk of overfitting. Therefore, parameter selection is crucial for faster model construction, improved accuracy, and enhanced interpretability. In this study, a stepwise regression approach was used to select parameters and develop a model for estimating the AGB in coniferous forests.

Stepwise regression, a widely used method for parameter reduction and regression [31], was compared with the RF-AGA model in this study. The goal was to determine which model achieved better parameter reduction and higher accuracy when fitting parameters for high-dimensional data in the dual-frequency band.

The present study aimed to address the following two questions using the stepwise regression algorithm:

(1) In the face of high-dimensional data, can the stepwise regression algorithm effectively reduce parameters and construct a high-precision and robust model?

(2) Based on the feature selection of prior knowledge (forest scattering mechanisms and sensitivity analysis), does the inclusion of artificially selected parameters in the stepwise regression equation enhance the ability of stepwise regression to estimate forest AGB?

*3.3. Model Validation*

Given the limited number of sample plots (50) employed in this study, we employed leave-one-out cross validation (LOOCV) for model testing. This method entails iteratively testing the model on the remaining plots, excluding one plot at a time, thereby enabling a comprehensive evaluation of its performance. Despite the constrained sample size, this approach offers a dependable means of assessing the model's predictive capabilities within the context of this study.

The model predictions were evaluated using several metrics, namely, the coefficient of determination ($R^2$), root mean-squared error (RMSE), mean absolute error (MAE), and bias (Equations (4)–(7)). These metrics were employed to assess the consistency between the field-estimated and model-predicted AGB [46].

$$R^2 = 1 - \left( \sum_{i=1}^{n} \frac{(Y_i - y_i)^2}{\sum_{i=1}^{n}(Y_i - \overline{y}_i)^2} \right) \tag{4}$$

$$RMSE = \sqrt{\sum_{i=1}^{n} \frac{(Y_i - y_i)^2}{n}} \tag{5}$$

$$MAE = \frac{\sum_{i=1}^{n} |Y_i - y_i|}{n} \tag{6}$$

$$Bias = \sum_{i=1}^{n} \frac{1}{n}(Y_i - y_i) \tag{7}$$

where $Y_i$, $y_i$, $\overline{y}_i$, and n represents the predicted AGB value, field AGB value, average value of true AGB value, and the sample size, respectively.

## 4. Results

*4.1. Result of Dominant Scattering Mechanism in C-Band and L-Band Data*

In the method of three-component decomposition, the dominant scattering mechanism of the L-band and C-band in the *Larix principis-rupprechtii Mayr* forest was volume scattering (the proportion of volume scattering for each decomposition method in the L-band and C-band ranges from 44–86% and 40–72%, respectively), followed by surface scattering, and the smallest proportion was double-bounce scattering (except for the VanZyl3 component decomposition result in the C-band) (Figure 3). Regardless of the three decomposition method mentioned in this study, the proportion of volume scattering and double-bounce scattering in the C-band was smaller than in the L-band, and the proportion of surface scattering in the C-band was larger than that in L-band. In the C-band, the surface scattering mechanism was the dominant mechanism (accounting for 47%) in the VanZyl3 component decomposition results, followed by the volume scattering mechanism (accounting for 40%) and the double-bounce scattering mechanism (accounting for 13%). This is because the VanZyl3 component decomposition method improves the overestimation of volume scattering in the Freeman three-component decomposition method [41], which reduces the proportion of volume scattering in the C-band in the VanZyl3 component and increases the proportion of surface scattering, making the VanZyl3 component decomposition results different from the other decomposition methods.

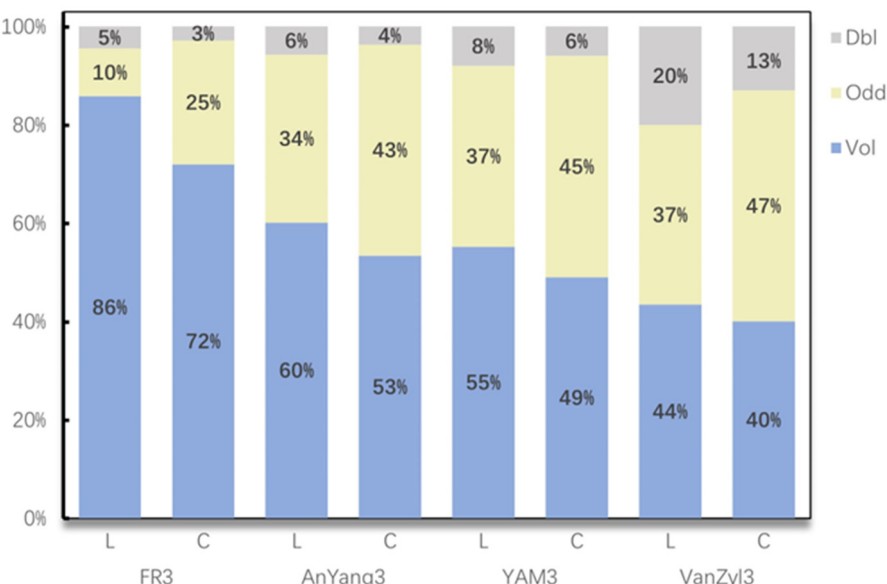

**Figure 3.** The proportion of each scattering mechanism of the C-band and L-band in the *Larix principis-rupprechtii Mayr* forest. Blue represents volume scattering, yellow represents surface scattering, and gray represents double-bounce scattering. FR3, AnYang3, YAM3, and VanZyl3 represent Freeman3 component decomposition, An-Yang3 component, AnYang3 component, Yamaguchi3 component, and Vanzyl3 component decomposition methods, respectively.

### 4.2. Result of Sensitivity Analysis

Among the backscatter coefficients of the C-band, only the backscatter coefficients of the cross-polarization channels (HV and VH) were more sensitive to the change in the AGB (Table 1).

**Table 1.** Correlation between the C-band backscatter coefficient and forest AGB.

| Parameter | Pearson Coefficient | Parameter | Pearson Coefficient |
| --- | --- | --- | --- |
| HH | 0.11 | VV/X | 0.33 ** |
| HV | 0.49 ** | HH + X | 0.36 ** |
| VH | 0.47 ** | VV + X | 0.3 * |
| VV | −0.01 | HH-X | −0.42 ** |
| (HV + VH)/2 | 0.48 ** | X-VV | 0.46 ** |
| HH/VV | −0.16 | HH-VV-X | 0.35 ** |
| HH/X | 0.28 * | VV * X | −0.24 |

Note: ** means significant at the 0.01 level, * means significant at the 0.05 level.

In the L-band, the backscatter coefficients of HV and VH were sensitive to changes in the forest biomass (Table 2), and the backscatter coefficients of the co-polar channel (HH) were also sensitive to changes in the forest biomass (R = 0.6). Among the backscatter coefficients of GF-3 and ALOS-2 and their combinations, the highest correlation with AGB was the HV polarization channel (R = 0.49, R = 0.68). The sensitivity between backscatter and forest AGB depends on the frequency, polarization, and angle of incidence of the SAR data [47]. In the combination of C-band backscatter coefficients, (HV + VH)/2 had the highest correlation with AGB (R = 0.48), but the combination of backscatter coefficients did not significantly improve the correlation with AGB, and the strongest correlation was the backscatter coefficient for the HV polarized channel. Among the combinations of backscatter coefficients in the L-band, HH × VV × X (X = (HV + VH)/2) had the highest correlation with the forest AGB (R = 0.7). The reason for the highest correlation of this combination is that this combination combined the backscatter information of four channels, which can contain more information about the interaction with AGB. The backscattering of

the HV and VH channels in the C-band was almost entirely from the volume scattering of the forest in the canopy, and the backscattering coefficient was highly dependent on the parameters in the canopy [48]. The backscattering of the L-band HV, VH, and HH channels basically came from the scattering in the canopy and trunks (the contribution of both the twig and needles was deemed insignificant) [21,22].

**Table 2.** Correlation analysis between the L-band backscatter coefficient and forest aboveground biomass.

| Parameter | Pearson Coefficient | Parameter | Pearson Coefficient | Parameter | Pearson Coefficient |
|---|---|---|---|---|---|
| VV | 0.49 ** | VV + X/HH | 0.36 ** | HH-X | −0.49 ** |
| VH | 0.66 ** | HH + VV + X/HH | 0.46 ** | X-VV | 0.56 ** |
| HV | 0.68 ** | HH + VV + X/VV | 0.46 ** | HH-VV-X | 0.53 ** |
| HH | 0.6 ** | HH + VV | 0.49 ** | HH × VV | −0.38 ** |
| X | 0.67 ** | HH + X | 0.64 ** | HH × VV | −0.68 ** |
| HH/VV | −0.15 | HH + VV + X | 0.6 ** | HH × VV × X | 0.7 ** |
| VV/X | 0.24 | VV + X | 0.6 ** | VV × X | −0.64 ** |

Note: ** means significant at the 0.01 level (X = $(\sigma_{VH} + \sigma_{HV})/2$).

In the C-band, the volume scattering component exhibited the strongest correlation with the forest AGB across the various decomposition methods, followed by the surface scattering component, while the secondary scattering component showed the lowest correlation with biomass (Table 3). Specifically, the volume scattering components from Freeman3, Anyang4, Yamaguchi4, and Pauli3 were found to be sensitive to changes in the forest AGB, with the volume scattering component from Freeman3 exhibiting the highest correlation with an R value of 0.38. Additionally, the C-band radar vegetation index (RVI) demonstrated a higher sensitivity to changes in the AGB with a correlation coefficient of 0.36.

**Table 3.** Correlation analysis between the polarization decomposition parameters in the C-band and aboveground biomass.

| Parameter | Pearson Coefficient | Parameter | Pearson Coefficient | Parameter | Pearson Coefficient |
|---|---|---|---|---|---|
| F2_V | 0.38 ** | lg(AY3_V) | 0.31 ** | Van_lgO/ | 0.33 ** |
| AY4_V | 0.36 ** | LgAY3o/(lgd * lgv) | 0.33 ** | Y3_O | 0.26 * |
| AY4_D | 0.19 | F3_V | 0.23 * | Y3_D | 0.23 |
| Y4_D | 0.2 | lg(F3_Vl) | 0.31 | Y3_V | 0.23 |
| Y4_V | 0.35 ** | F3D * F3V | -0.33 ** | lg(Y3_v) | 0.31 * |
| Pauli_D | 0.32 ** | Van3_O | 0.24 * | Y3_O/v * d | 0.32 * |
| Pauli_V | 0.38 ** | Van3_V | 0.23 * | RVI_Freeman | 0.36 ** |
| AY3_O | 0.18 | lgVan_O | 0.3 ** | | |
| AY3_V | 0.22 * | lgVan_V | 0.31 ** | | |

Note: ** indicates significance at the 0.01 level, * indicates significance at the 0.05 level. F2, F3, AY4, AY3, Y3, Y4, and Van3 represent the Freeman two-component decomposition, Freeman3 component decomposition, AnYang3 component, AnYang4 component, Yamaguchi3 component, Yamaguchi4 component, and Vanzyl3 component decomposition methods, respectively. V, D, and O represent the volume scattering, double-bounce scattering, and surface scattering component, respectively.

Comparatively, the correlation between each scattering component of the L-band decomposition methods and forest AGB was higher than that observed for the C-band decomposition methods. However, the volume scattering components in both the C-band and L-band exhibited better sensitivity to forest AGB compared to other scattering components.

Table 4 shows the sensitivity analysis between the polarization decomposition parameters of the L-band and forest AGB. In the L-band, the volume scattering components (vol) from the Freeman 2, Freeman3, AnYang3, and Yamaguchi3 decomposition methods exhibited the strongest correlation with forest AGB. Specifically, the volume scattering component of the Freeman2 method in the L-band showed a higher correlation with the biomass compared to Freeman3 and Yamaguchi3, with an R value reaching 0.43.

**Table 4.** Correlation analysis between the L-band polarization decomposition parameters and aboveground biomass.

| Parameter | Pearson Coefficient | Parameter | Pearson Coefficient | Parameter | Pearson Coefficient |
|---|---|---|---|---|---|
| F2V | 0.43 ** | Entropy sh | 0.34 ** | lgY3D * lgY3O | −0.36 ** |
| lg(F2V) | 0.46 * | entropy1 | 0.34 ** | Van3_V | 0.39 ** |
| AY3V | 0.39 ** | entropy4 | 0.42 ** | Van3_O | 0.33 ** |
| F3V | 0.34 * | entropy5 | 0.44 ** | Van3_D | 0.43 ** |
| lg(F3V) | 0.42 ** | anisotro1 | 0.31 * | lg(Van3_V) | 0.44 ** |
| lg(F3 D) | 0.31 * | Y3_V | 0.4 ** | lg(Van3_Odd) | 0.37 ** |
| lgF3V * lgF3O | −0.41 ** | Y3_D | 0.28 * | lg(Van3_D) | 0.36 ** |
| (lgF3d/lgF3v * lgF3O) | −0.36 * | lg(Y3_V) | 0.45 ** | lg(Van3 O) * lg(Van3_D) | −0.43 ** |
| lgF3V * lgF3D | −0.42 ** | lg(Y3_D) | 0.36 ** | AnY3_V | 0.33 * |

Note: ** indicates significance at the 0.01 level, * indicates significance at the 0.05 level. F2, F3, AY3, Y3,Y4, and Van3 represent the Freeman two-component decomposition, Freeman3 component decomposition, AnYang3 component, Yamaguchi3 component, Yamaguchi4 component, and Vanzyl3 component decomposition methods, respectively. V, D, and O represent the volume scattering, double-bounce scattering, and surface scattering component, respectively.

For the L-band VanZyl method, all three components demonstrated a significant correlation with the forest AGB, indicating their sensitivity to changes in the AGB. Among these components, the secondary scattering component exhibited the highest sensitivity to biological changes, with an R value as high as 0.43, followed by the volume scattering component (R = 0.39), and finally, the surface scattering component (R = 0.33). Upon applying logarithmic transformations and combining the polarization decomposition parameters, their correlation with AGB improved to varying degrees. After the logarithmic change, the volume scattering parameter of the Freeman2 method exhibited the highest correlation with AGB (R = 0.46) and passed the significance test. Furthermore, the sensitivity of the L-band backscattering coefficient to AGB was found to be stronger than that of the polarization decomposition parameters.

*4.3. Results of Feature Selection*

The results of selecting the backscatter coefficients based on correlation analysis were as follows. In the L-band, a total of 13 backscatter coefficients and their combinations were retained: HH, HV, VH, VV, (HV + VH)/2, HH + X, HH + VV + X, VV + X, X-VV, HH-VV-X, HH × X HH × VV × X VV × X (X = (HV + VH)/2). In the C-band, a total of 11 backscatter coefficients and their combinations were retained: HH, HV, VH, VV, (HV + VH)/2, HH/X, VV/X, HH + X, HH-X, X-VV, and HH-VV-X.

Based on the principle of selecting the dominant scattering mechanism, the results of selecting the polarization decomposition parameters were as follows. The dominant scattering mechanisms were retained for each decomposition method in the C-band and L-band, while the secondary scattering parameters of the Freeman3 decomposition method in both the C-band and L-band were removed. Additionally, the helix scattering component in the Anyang4 and Yamaguchi4 decomposition methods as well as the secondary scattering parameters in the C-band Anyang3 decomposition method were excluded.

The results of selecting the polarization decomposition parameters based on correlation analysis were consistent with the results of the screening parameters based on the dominant scattering mechanism, and the parameters that dominated the scattering were also biomass sensitive factors, which passed the significance test. A total of 91 parameters were selected based on prior knowledge (the dominant scattering mechanism and sensitivity analysis) including 51 C-band parameters and 40 L-band parameters (Table 5).

**Table 5.** Results of the C-band and L-band feature selection based on the scattering mechanisms and sensitivity analysis.

| Parameter/Method | Number of Parameters (C) | Number of Parameters (L) |
|---|---|---|
| Backscatter coefficients and their combinations | 11 | 13 |
| Freeman2 component | 2 | 2 |
| Freeman3 component | 3 | 4 |
| AnYang3 component | 4 | 4 |
| AnYang4 component | 3 | |
| Yamaguchi3 component | 9 | 7 |
| Yamaguchi4 component | 3 | |
| VanZyl3 component | 9 | 5 |
| Pauli3 component | 6 | |
| H/A/alpha eigenvalue set decomposition | | 5 |
| RVI | 1 | |

Note: C and L mean the C-band and L-band, respectively. RVI represents the radar vegetation index.

### 4.4. Results of AGB Model and Validation

The improved forest aboveground biomass model was created by combining the non-parametric model (random forest adaptive genetic algorithm model) and feature selection based on the dominant scattering mechanism and sensitivity analysis, utilizing the GF-3 data in the C-band and ALOS-2 data in the L-band. In order to validate the potential of using the combined C-band and L-band for estimating biomass, which is superior to using the C-band or L-band alone, we incorporated data from the C-band, L-band, and the combined C-band with L-band into the non-parametric and stepwise regression models. To validate the potential of improving the aboveground biomass (AGB) estimation in the model by first selecting parameters based on the scattering mechanism and sensitivity analysis, we separately input the data from the combined C-band and L-band as well as the data selected based on the parameters of the C-band and L-band into the non-parametric and stepwise regression models.

Figure 4 shows the validation results of AGB estimation using the random forest adaptive genetic algorithm (RF-AGA) model and stepwise regression model. The results indicate that the RF-AGA model that selected parameters through the dominated scattering mechanism and sensitivity analysis by incorporating 91 parameters of both the C-band and L-band data achieved the highest accuracy in estimating the forest AGB, with the lowest root mean square error (RMSE) of 10.42 t/ha and the best fit ($R^2$ = 0.93). Moreover, this model demonstrated the most effective parameter simplification, with the final model including only eight parameters representing the Yamaguchi3 volume scattering component, entropy 1 of H/A/alpha eigenvalue set decomposition, lg(Freeman3_vol), VanZly3_Dbl for the L-band and $\sigma_{vv+(vh+hv)/2}$, $\sigma_{vv\times(vh+hv)/2}$, Freeman2_vol, and VanZyl_Odd for the C-band. Compared to models that solely estimated the AGB using either the C-band or L-band parameters, the model combining the C-band and L-band parameters exhibited superior estimation performance within the biomass range of 0–200 t/ha. At biomass levels exceeding 200 t/ha, the model underestimated AGB, although no significant saturation phenomenon was observed. The inclusion of 91 parameters from the C-band and L-band, selected based on prior knowledge (dominated scattering mechanism and sensitivity analysis), into the stepwise regression model ($R^2$ = 0.72) for estimating the forest AGB yielded superior results compared to the stepwise regression model without feature selection (176 parameters from the C-band and L-band), particularly for high biomass levels exceeding 150 t/ha. Overall, feature selection with scattering mechanisms and sensitivity analyses improved the accuracy of the biomass estimates for both the linear and non-parametric models using C-and L-band data.

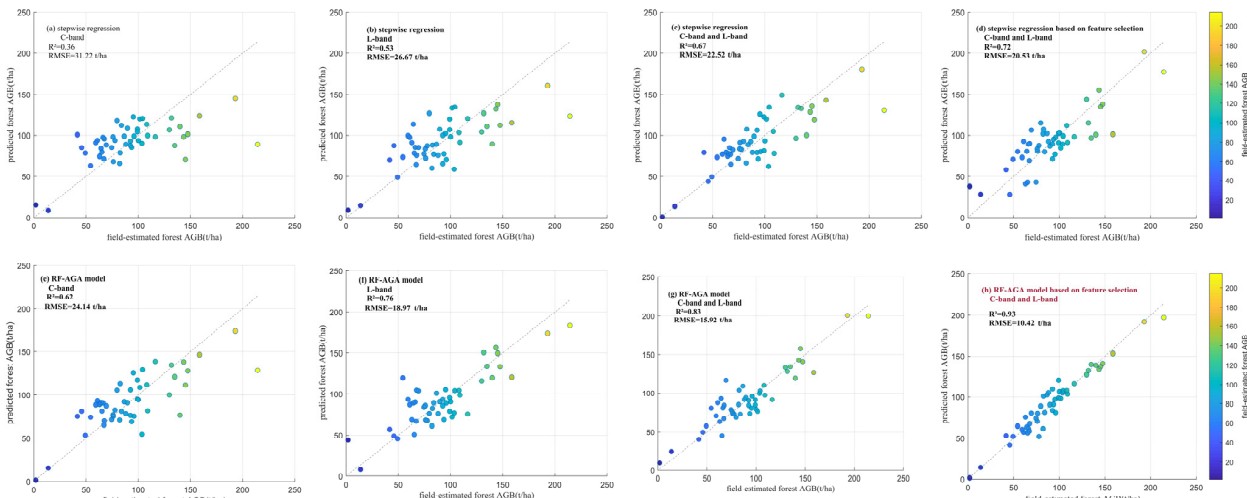

**Figure 4.** The validation results of AGB estimation by the non-parametric model and stepwise regression model. (**a**–**d**) correspond to the stepwise regression results of the backscatter coefficients and polarimetric decomposition parameters for the GF-3 data, ALOS-2 data, combined GF-3 data and ALOS-2 data, and the GF-3 data and ALOS-2 data with preselected parameters based on feature selection, respectively. (**e**–**h**) represent the results of the non-parametric model (RF-AGA) estimation using the C-band, L-band, C-band combined with L-band, and the C-band and L-band based on feature selection, respectively. The 1:1 line is shown in grey.

The model that utilized C-band and L-band data, considering the backscattering and polarization parameters, demonstrated superior performance in estimating the biomass compared to solely using the C-band or L-band data, and can effectively resolve the problem of significant overestimation observed when using individual C-band or L-band data within the biomass range of 50–150 t/ha as well as the issue of notable underestimation within the biomass range of 150–220 t/ha.

This study concluded that the non-parametric (RF-AGA) model based on feature selection using the C-and L-band data was the best model when compared with the stepwise regression model and the RF-AGA model, which was not selected with the dominant scattering mechanisms and sensitivity analysis. Figure 5 shows the forest biomass map using the RF-AGA model based on feature selection.

The results of model validation (Table 6) indicates that the estimation model for AGB using the C-band and L-band demonstrated higher levels of fitting and accuracy compared to using either the C-band or L-band models separately.

In general, the predicted aboveground biomass (AGB) values of the models were consistently higher than the ground-based measured AGB values within the biomass range of 0–100 t/ha, resulting in positive bias values. However, it should be noted that for the RF-AGA algorithm's CL model, the individual predicted values were lower than the ground-based measured AGB values within the 0–50 t/ha AGB range, yielding negative bias values. For the biomass range of 100–220 t/ha, the predicted values of the stepwise regression model incorporating backscattering and polarization decomposition as well as the RF-AGA algorithm's models were consistently lower than the ground-based measured biomass values. An exception to this was the RF-AGA model using the C- and L-band data, which slightly overestimated the ground-based measured values within the 100–150 t/ha biomass range with a bias of 0.08. At the level of 200–220 t/ha, the model bias value was higher than other biomass levels, the reason being that there were less training samples, resulting in a poor fitting effect (Table 7).

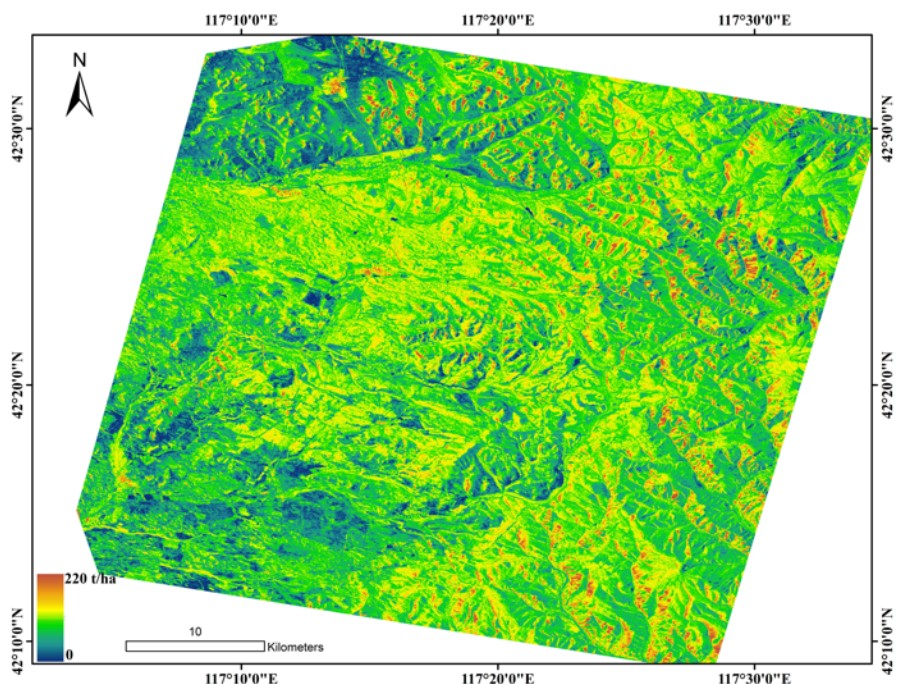

**Figure 5.** The map of AGB estimation by the RF-AGA model based on feature selection.

**Table 6.** The results of the leave-one-out crossover validation model.

| Model | R$^2$ | RMSE | MAE |
|---|---|---|---|
| Model1 | 0.36 | 31.22 | 22.23 |
| Model2 | 0.53 | 26.67 | 20.03 |
| Model3 | 0.67 | 22.52 | 17.75 |
| Model4 | 0.72 | 20.53 | 15.26 |
| Model5 | 0.62 | 24.14 | 18.61 |
| Model6 | 0.76 | 18.97 | 15.59 |
| Model7 | 0.83 | 15.92 | 12.72 |
| Model8 | 0.93 | 10.42 | 8.18 |

Models 1–4 represent stepwise regression models for the backscattering coefficient and polarization decomposition parameters in the C-band, L-band, and the combined C-band and L-band, and the C-band and L-band based on feature selection, respectively. Models 5–8 correspond to the RF-AGA models for the C-band, L-band, combined C-band and L-band, and the C-band and L-band based on feature selection, respectively.

**Table 7.** Results of the model bias.

| AGB Level. | M1 | M2 | M3 | M4 | M5 | M6 | M7 | M8 |
|---|---|---|---|---|---|---|---|---|
| 0–50 t/ha | 14.4 | 3.2 | 8.1 | 4.5 | 6.2 | 6.6 | 2.1 | −1.0 |
| 50–100 t/ha | 7.4 | 5.9 | 2.2 | 2.4 | 5.9 | 3.2 | 1.2 | 1.1 |
| 100–150 t/ha | −8.5 | −5.7 | −2.7 | −2.5 | −9.5 | −6.8 | −0.9 | 0.1 |
| 150–200 t/ha | −31.7 | −7.0 | −3.0 | −12.0 | −16.7 | −8.6 | −5.0 | −4.7 |
| 200–220 t/ha | −73.1 | −85.4 | −58.7 | −20.2 | −21.6 | −8.8 | −18.6 | −13.8 |

Note: M1–M4 represent stepwise regression models for the backscattering coefficient and polarization decomposition parameters in the C-band, L-band, and the combined C-band and L-band, and prior knowledge-selected parameters for the C-band and L-band, respectively. M5–8 correspond to the RF-AGA models for the C-band, L-band, combined C-band and L-band, and prior knowledge-selected parameters for the C-band and L-band, respectively.

These results indicate that the models incorporating prior knowledge-selected feature exhibited smaller biases compared to models without parameter selection. This suggests that the combination of parameter selection based on scattering mechanisms and sensitivity analysis, followed by the utilization of machine learning RF-AGA algorithm, leads to improved fitting performance for aboveground biomass estimation.

## 5. Discussion

This paper proposed a research methodology for the estimation of forest AGB using C- and L-band SAR data that incorporates feature selection based on dominant scattering mechanisms and sensitivity analysis results using a non-parametric (RF-AGA) model. By introducing feature selection and a non-parametric model, we mitigated the risk of selecting the local optima and redundant parameters and achieved a robust and highly accurate AGB model of a coniferous forest.

### 5.1. Advantage of Combination C- and L-Band Data on AGB Estimation

The advantage of C-band (GF-3 data) and L-band (ALOS-2 data) in coniferous forests primarily encompasses three aspects: the scattering mechanisms, backscatter coefficients, and the analysis of the polarization decomposition parameters.

### 5.1.1. Advantage of Dominant Scattering Mechanisms in C- and L-Bands

The advantage of combining C-band and L-band data includes the interaction information between the C-band and fine-scale branch and needle structures within the forest canopy as well as the interaction information between the L-band and branches situated within the canopy layer and trunks beneath the canopy.

In the C-band, the needles (leaves) and the upper elements of the canopy often contribute almost all of the signals [44,48]. The wavelength of the C-band is short (5.5 cm), and the size of the canopy is much larger than the length of the wavelength. Therefore, the C-band exhibits greater sensitivity to fine-scale structures within the canopy, whereas it is less sensitive to the main branches within the canopy compared to the L-band. The scattering mechanisms (volume scattering, surface scattering, and double-bounce scattering) of the C-band interaction with the forest are all concentrated in the interaction with the forest canopy [44,48]. Among them, the volume scattering of the C-band occurs on the secondary branch and small needles in the upper part of the canopy, the surface scattering occurs on the surface of the canopy of the dense forest or on the surface of the sparse forest, and the double-bounce scattering mainly occurs on the surface of the canopy among the thicker and larger branches.

The L-band has a longer wavelength (compared to the C-band) and has strong penetration. It is not as sensitive to the small structural changes in the canopy as the C-band, and is more connected to the main branches in the canopy. Additionally, the L-band exhibits the ability to penetrate the canopy, interacting with both the trunk and the ground. However, the scattering of the L-band in the coniferous forest of Saihanba in China mainly comes from the canopy (branches and trunks in the canopy). Because most of the forests in the Saihanba area are still in the stage of middle-aged and young forests, the dense canopy strongly reflects and attenuates SAR signals, inhibiting the reflection of SAR signals from the lower layers of the canopy. In the L-band, the volume scattering mainly represents the interaction with the branches and trunks in the canopy [22]. The double-bounce scattering represents the interaction between the trunk and the ground or between the canopy and the ground. The surface scattering represents the interaction between the L-band and the sparse forest ground as well as the interaction with the canopy surface of a dense forest. Because the L-band has strong penetrability, it can not only interact with the upper part of the tree canopy, but can also interact with the entire tree canopy. Therefore, regardless of what target decomposition method is based on the statistical results, the proportion of volume scattering in the L-band is more than the C-band. The proportion of double-bounce scattering in the L-band under different polarization decomposition methods was the least (5–20%). This is because the average DBH of all plots was 13.4 cm, which is much smaller than the wavelength of the L-band (23.6 cm), so the double-bounce scattering components of the trunk and the ground in the L-band account for less in young and middle-aged coniferous forests. This was the same as the result of Moghaddam et al. [48], but different from the results of Durden et al. [49], where the double-bounce scattering and surface scattering contributions of the L-band in the tropical forest were almost zero. This discrepancy may be attributed to the canopy density of the original tropical forest being much higher than that

of coniferous forest in northern China. The dense canopy reflects and attenuates the SAR signal to a large extent, so the L-band cannot produce secondary scattering and surface scattering contributions from the tree trunks and rough ground below the canopy.

The nature of the interaction of the SAR signal with the forest makes SAR sensitive to the geometric structure of the forest, therefore SAR is also very sensitive to the biomass (biophysical characteristics) of the forest [50]. Notably, the L-band exhibits a greater extent of double-bounce scattering compared to the C-band. The double-bounce scattering primarily arises from interactions involving the trunks and the ground, with the trunks containing a significant portion of the tree biomass. Combining the results of dominant scattering mechanisms in forests conducted in this study and the same results in previous studies [40,43,48], the scattering characteristics of the C-band and L-band in a *Larix principis-rupprechtii Mayr* forest in North China are summarized as shown in Table 8.

**Table 8.** Scattering conditions of the C-band and L-band in forests.

| Scattering Type | Scattering Mechanism | Scattering Situation of C-Band in Forest | Scattering Situation of L-Band in Forest |
|---|---|---|---|
| Volume scattering | A cloud of a randomly of the oriented dipole | Small structures such as needles in the canopy or sparse forest land–canopy | The main branches in the canopy or ground–canopy |
| Surface scattering | Moderately rough surface | Density canopy surface, rough surface of sparse woodland or bare woodland | Rough ground, young forest stand |
| Double-bounce scattering | Modeled by scattering from a dihedral corner reflector | With major branches in the canopy (rarely) | Trunk–ground |
| Helix scattering | Left or right circular polarization states | Almost nonexistent | Almost nonexistent |

5.1.2. Advantage of Backscatter in the C- and L-Bands

The advantage of combining C-band and L-band backscatter information is that the C-band and L-band backscatter coefficients have different sensitivities to biomass and contain information of different forest components to improve the potential of SAR data to estimate biomass. The highest correlation with AGB is the HV polarization channel in the C-band and L-band, which is consistent with the results in previous studies [19,22,51]. Each polarization channel in the C-band has a lower backscatter coefficient value than the corresponding scattering channel in the L-band (Figure 6). The backscatter coefficient values for the C-band and L-band cross-polarization channels (HV, VH) are generally lower than those of the co-polarization channels (HH, VV). This is because forests, with their complex canopy structure and randomly distributed foliage, predominantly exhibit volume scattering. This complex environment results in a strong depolarization phenomenon for the SAR signal of cross-polarization channels [52], hence the lower value of the backscatter coefficient for the cross-polarization channels. The higher HH channel backscattering coefficient in coniferous trees may be attributed to the more prevalent arrangement of dominant tree branches at a constant angle [45]. This can be attributed to the fact that HH polarization primarily interacts with the surface and trunk backscattering components, which capture a significant portion of the vertically elongated structures where most of the biomass is stored. In general, regardless of the SAR frequency, co-polarized signals (HH or VV) demonstrate higher backscattering intensities than cross-polarized signals (HV or VH) due to reduced scattering and speckle effects [44].

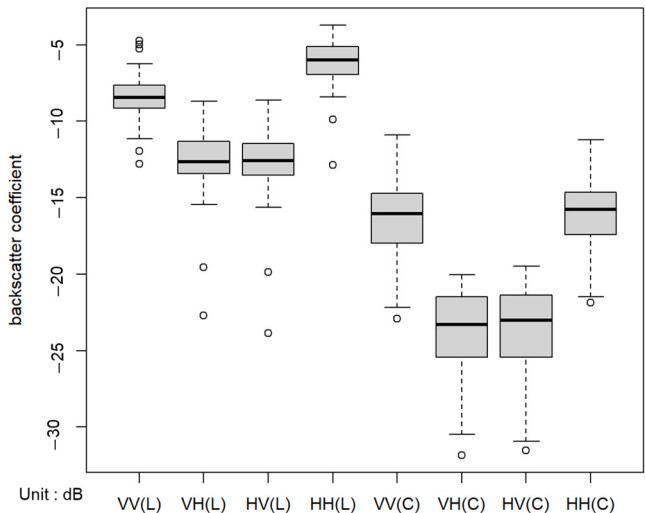

**Figure 6.** Boxplot of the C-band and L-band backscatter coefficients.

### 5.1.3. Advantage of Polarization Decomposition Parameters in the C- and L-Bands

For the C-band, the Pauli three-component decomposition method, the Yamaguchi three-component decomposition method, and the VanZyl three-component decomposition method are all suitable for estimating the forest biomass. The decomposition parameters are more sensitive to forest biomass. For the L-band, the VanZyl three-component decomposition method and the Yamaguchi three-component decomposition method are both suitable for estimating the forest biomass, and the Vanzyl3 component decomposition method is the most suitable target decomposition method for estimating the forest biomass. The parameters obtained by VanZyl3 component decomposition and the new constructed parameter combinations are all sensitive factors of biomass, and more sensitive to the change in the forest AGB than the parameters obtained by other methods.

### 5.2. Advantage of Feature Selection and Non-Parametric Model to Estimate AGB

The non-parametric model (random forest adaptive genetic algorithm), which can avoid the problem of the local optimum, employed in this study offers the advantage of performing feature selection and controlling the hyperparameters simultaneously on the input feature subsets. Because this study selected 176 backscatter coefficients and polarization decomposition parameters of the C-band and L-band and their respective combination parameters, there will be a phenomenon of parameter redundancy. Therefore, prior to incorporating SAR parameters into the random forest adaptive genetic algorithm (RF-AGA) model, a parameter selection process was conducted based on the dominant scattering mechanisms in forested areas and a sensitivity analysis of the C-band and L-band. This process involves removing redundant parameters that contain repetitive information as well as eliminating parameters with low sensitivity to biomass or those representing scattering components that do not exist or have a minimal presence in forests. The objective is to simplify the number of input parameters into the model, enhancing the optimization efficiency of the non-parametric model and improving the potential of biomass estimation.

This study demonstrated that models utilizing C-band and L-band data, based on prior knowledge for feature selection, outperformed models without parameter selection in estimating the biomass. The $R^2$ values for the stepwise regression model and the RF-AGA model increased from 0.67 to 0.72 and from 0.83 to 0.93, respectively. As an illustration, the results presented by Prakash et al. [3] revealed that the accuracy of estimating the AGB, achieved through the utilization of a random forest algorithm that did not incorporate feature selection based on prior knowledge and incorporated dual-frequency SAR data ($R^2 = 0.89$), fell below the accuracy ($R^2 = 0.93$) attained in the present study. Comparing our research findings with those of previous studies that used the RF algorithm to select parameters and utilized regression-based models to estimate the AGB, it is evident that our

proposed method achieved a higher predictive accuracy. For instance, while our model yielded an $R^2$ value of 0.72 using a stepwise regression model based on prior knowledge for feature selection, Zeng et al. [19] reported an $R^2$ value of $R^2$ = 0.562 using C-band and L-band data. Estimating AGB using both the C-band and L-band data surpassed those achieved by using C-band or L-band models alone, regardless of whether it is a stepwise regression model or a non-parametric model. This dual-frequency SAR model leverages the respective advantages of the C-band and L-band, considering a broader range of interactions between the forest biomass components and SAR signals. These results align with previous findings by Englhart et al. [15], highlighting the enhanced estimation accuracy and saturation point for AGB achieved through the integration of high-frequency and low-frequency SAR data. Furthermore, no apparent saturation point was observed, indicating the sensitivity of the selected parameters to variations in AGB and their practical physical significance in the scattering mechanism. Conversely, the selected-out parameters contributed minimally to the model, simplifying the parameter set while improving the accuracy, saturation point, and efficiency of AGB estimation, particularly for the *Larix principis-rupperchtii Mayr* forest AGB. The estimation of AGB using SAR parameters and their combination extracted solely from the L-band demonstrated higher accuracy and a better fit compared to using the C-band alone. This can be attributed to the strong penetration capability of the L-band, enabling it to interact with the larger trunk containing a significant portion of the biomass, along with the main branches. Conversely, the C-band has limited penetration capability and primarily interacts with the small branches storing a smaller biomass [53]. These results were the same as the findings of Prakash et al. [3], but differed from the research results of Zeng et al. [19], who reported a better estimation of AGB using the backscattering coefficient and polarization decomposition parameters of the C-band compared to the L-band. The disparity may be attributed to site 1 in the work by Zeng et al., which was characterized by a low biomass level and sparse trees, favoring the C-band's performance in AGB estimation.

## 6. Conclusions

In this study, the backscattering coefficients and polarization decomposition parameters of the C- and L-band SAR data were used to apply the non-parametric model (RF-AGA) based on feature selection of the dominant scattering mechanism and sensitivity analysis to improve the accuracy and efficiency of dual-frequency SAR estimation of forest AGB under the condition of limited sample plots. The dominant scattering mechanism of the L-band and C-band in the Freman3 component, AnYang3 component, and Yamaguchi3 component was volume scattering, followed by surface scattering, and the proportion of double-bounce scattering was the smallest. In the VanZyl three-component, the C-band dominant scattering mechanism was surface scattering, followed by volume scattering, and double-bounce scattering accounted for the smallest proportion. The L-band estimation of the scattering mechanism of the North China larch has the advantage that it can fully interact with the main branches and trunks of the tree canopy. The backscattering coefficients of the L-band HV, VH, and HH three polarization channels and the parameters of the Vanzyl3 component decomposition method are more suitable for estimating the aboveground biomass. The advantage of the C-band is that it can fully interact with the needles and small branches in the canopy. The HV VH backscatter coefficient, the Pauli three-component decomposition method, and the VanZyl3 component decomposition method are more suitable for the C-band estimation of the forest aboveground biomass. Combining the estimation advantages of the C-band and L-band can more comprehensively reflect the information of the fine structures, main branches, and trunks in the forest canopy, and can better improve the saturation point of estimating the forest AGB. The fitting degree and accuracy of the biomass model estimated by using the machine learning RE-AGA algorithm selected by prior knowledge (forest-dominated scattering mechanism and biomass sensitivity analysis) of the C-band and L-band data were better than those of SAR parameters without. After selection, the fitting effect of directly entering the

model was good, and it was verified by the leave-one-out cross validation method where RMSE = 10.42 t/ha and $R^2$ = 0.93. However, this study only used 50 sample plots. Although the results are convincing, more observational data are needed to verify the results. This study introduced a research method that was applied to coniferous plantations in northern China, with the potential for adaptation to other forest types, pending further validation and analysis. According to the specific forest type and combining the advantages of the scattering mechanism of different frequency bands in the forest and the advantages of the forest biomass sensitive factors, analysis was conducted to improve the potential of the non-parametric model to estimate the biomass of different forest types. This research method can also be applied to regional-scale biomass estimation research in different climate zones and regions in the future, and the dual-band SAR data can be extended to multi-band SAR data by selecting the appropriate frequency band data according to the forest biomass level and forest density in the study area. Combining the feature selection and non-parametric model of dual-frequency SAR to estimate temperate forest biomass can lay the foundation for the quantitative estimation of regional forest biomass and also become an important part of the global carbon cycle model and climate change research.

**Author Contributions:** Conceptualization, W.F. and Y.H.; Methodology, Y.H.; Software, Y.H., G.W. and Z.L.; Validation, Y.H.; Formal analysis, Y.H.; Investigation, W.F. and Y.H.; Resources, W.F.; Data curation, Y.N., Y.H. and Z.L.; Writing—original draft preparation, Y.H.; Writing—review and editing, Y.H. and W.F; Visualization, Y.H. and Y.N. All authors have read and agreed to the published version of the manuscript.

**Funding:** This research was funded by the National Natural Science Foundation of China (contract. no. 31971654, funder: National Natural Science Foundation of China) and the Civil Aerospace Technology Advance Research Project (contract no. D040114).

**Data Availability Statement:** ALOS-2 Data was obtained from the Japan Aerospace Exploration Agency and are available at https://www.eorc.jaxa.jp/ALOS-2/en/about/palsar2.htm.

**Conflicts of Interest:** The authors declare no conflict of interest.

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
