# Peer review of "Improving the Potential of Coniferous Forest Aboveground Biomass Estimation by Integrating C- and L-Band SAR Data with Feature Selection and Non-Parametric Model"

_remotesensing, doi:10.3390/rs15174194_

Round 1

Reviewer 1 Report

The study deals with the estimation of coniferous forest above-ground biomass by integrating C- and L-band SAR observations through feature selection and non-parametric model. The manuscript was well written and organized. Several advices presented here for improving the manuscript for publication in remote sensing.

1.      Please add the detail for terrain radiometric correction of PALSAR and GF-3.

2.      Line 294 to 301 are suggested to move to introduction section.

3.      Cross-validation are suggested for results and model validation.

4.      Please add some detail of RF-AGA and then the readers are no need to read other paper.

5.      ‘Application in the future work’ is suggested to move to conclusion section.

6.      Table5, Table6, Table7, and Table 8 and related contents are suggested to move to results section.  

Minor editing of English language required.

Author Response

Thank you for your suggestion. Please see the attachment.

Reviewer 2 Report

Study approach seems good in terms of attempting two different frequencies of SAR and using non-parametric model to infer the scattering mechanisms of conifers. RF-AGA model with feature selection to infer AGB estimation performance seems appropriate. I suggest a comparison with either optical or single frequency SAR based recent findings to include for better comprehension of the merits of this approach.

Minor language editing at some places in overall text should improve it. Though I am not pointing out specific places, but recommend to have a thorough check.

Author Response

(The authors gave the same response as above.)

Reviewer 3 Report

This paper addresses forest biomass estimation using various parameters of SAR data. The authors quantitatively show which scattering components are effective for forest biomass estimation, which is meaningful information for concerned researchers. Therefore, I believe this paper is worth publishing. However, I recommend the following minor revisions.

1. [line 36] Typo: CO2 -> 2 as subscripts (and elsewhere)

2. [line 89] Although aboveground biomass is abbreviated here for the first time as AGB, it should be abbreviated where the term first appears. And after the abbreviation is described, the non-abbreviated term should not be described.

3. [line 124] Please describe the area or spatial scale of the study site.

4. [line 140] The display range of Figures 1(b) and 1(c) should be consistent to make it easier to identify.

5. [line 144] Please describe the year in which the 50 plots were measured. And also describe whether that year is close enough to the year of observation of the SAR image used.

6. [line 373] Typo: R2 -> 2 as superscript

7. [line 400] The R2 and RMSE shown in Figure 3 are different from those in Table 1. Perhaps, Figure 3 is not the result of leave-one-out cross validation, but the result of applying the estimation model to the training data. Readers may consider such scatter plots as results of accuracy validation. I would recommend to create scatter plots based on the results of validation.

8. [lines 563-565] I do not think it is feasible to derive the findings in Table 3 from the results of this study alone. I would recommend to describe, for example, the results of this study are consistent with the findings of previous studies, which can be summarized as shown in Table 3.

9. [line 587] Show the unit of vertical axis in Figure 6.

10. [line 613] Why are some polarization descriptions lower case (e.g., hh)?

11. I feel that the figures, tables and accompanying text in the Discussion section would be more appropriately placed in the Results section. Please consider the structure of the manuscript.

Author Response

(The authors gave the same response as above.)

Round 2

Reviewer 1 Report

The author revised the manuscript according to the comments. I agree to accept as the current version.